# Eco-Friendly Synthesis of an Oxovanadium(IV)-*bis*(abietate) Complex with Antimicrobial Action

**DOI:** 10.3390/molecules27196679

**Published:** 2022-10-07

**Authors:** Aline B. Schons, Jamille S. Correa, Patricia Appelt, Daiane Meneguzzi, Mário A. A. Cunha, Carla Bittencourt, Henrique E. Toma, Fauze J. Anaissi

**Affiliations:** 1Department of Chemistry, Universidade Estadual do Centro-Oeste, UNICENTRO, Alameda Elio Antonio Dalla Vecchia, 838, Guarapuava 85040-167, PR, Brazil; 2Department of Chemistry, Universidade Tecnológica Federal do Paraná, UTFPR, Via do Conhecimento, KM 01, Fraron, Pato Branco 85503-390, PR, Brazil; 3Department of Chemistry, University of Mons, Place du Parc 23, 7000 Mons, Belgium; 4Institute of Chemistry, University of Sao Paulo, São Paulo 05508-000, SP, Brazil

**Keywords:** green chemistry, abietate, vanadium complex, pigment, antifungal, antibacterial

## Abstract

The search for less expensive and viable products is always one of the challenges for research development. Commonly, the synthesis of coordination compounds involves expensive ligands, through expensive and low-yield routes, in addition to generating toxic and unusable residues. In this work, the organic ligand used is derived from the resin of a reforestation tree, *Pinus elliottii var.* *elliottii*. The synthesis method used Pinus resin and an aqueous solution of vanadium(III) chloride at a temperature of 80 °C. The procedure does not involve organic solvents and does not generate toxic residues, thus imparting the complex formation reaction a green chemistry character. The synthesis resulted in an unprecedented oxovanadium(IV)-*bis*(abietate) complex, which was characterized by mass spectrometry (MS), chemical analysis (CHN), vibrational (FTIR) and electronic spectra (VISIBLE), X-ray diffraction (XRD), and thermal analysis (TG/DTA). Colorimetric studies were performed according to the CIELAB color space. The structural formula found, consisted of a complex containing two abietate ligands, [VO(C_20_H_29_O_2_)_2_]. The VO(IV)-*bis*(abietate) complex was applied against microorganisms and showed promising results in antibacterial and antifungal activity. The best result of inhibitory action was against the strains of Gram-positive bacteria *S. aureus* and *L. monocytogenes*, with minimum inhibitory concentration (MIC) values of 62.5 and 125 μmol L^−1^, respectively. For Gram-negative strains the results were 500 μmol L^−1^ for *E. coli*; and 1000 μmol L^−1^ for *Salmonella enterica* Typhimurium. Antifungal activity was performed against *Candida albicans*, where the MIC was 15.62 μmol L^−1^, and for *C. tropicalis* it was 62.5 μmol L^−1^. According to the MFC analysis, the complex presented, in addition to the fungistatic action, a fungicidal action, as there was no growth of fungi on the plates tested. The results found for the tests demonstrate that the VO(IV)-*bis*(abietate) complex has great potential as an antimicrobial and mainly antifungal agent. In this way, the pigmented ink with antimicrobial activity could be used in environments with a potential risk of contamination, preventing the spread of microorganisms harmful to health.

## 1. Introduction

The resin is a part of the tree’s natural defense mechanism and is composed of diterpenic carboxylic acids, of which 47% is abietic acid [1,2,3]. Through the carboxylate group (COO^−^), the molecule readily acts as a ligand for the formation of stable carboxylate complexes. Carboxylic acids and their derivatives occupy an important place in science, as they have different coordination modes and medicinal characteristics; pharmaceutical activities, such as antimicrobial, analgesic, anti-inflammatory, and hypoglycemic agent; and also act as an intermediate product in the synthesis of drugs for the treatment of autoimmune diseases, among other activities [1,4].

Thus, the carboxylic group (COO^−^) has an important role in the preparation of metal complexes. For example, Lu et al. [5] report the synthesis of the metal-organic structures Ag(MOFs), including [Ag_2_(O-IPA)(H_2_O)(H_3_O)] and [Ag_5_(PYDC)_2_(OH)], with aromatic carboxylic acids, such as 5-hydroxyisophthalic acid (HO-H_2_IPA) and pyridine-3,5-dicarboxylic acid (H_2_PYDC). They showed excellent antimicrobial activities against Gram-positive and Gram-negative bacteria. Another example is the iron nanoparticle (NPs) compound [MOF-53(Fe)], synthesized using carboxylic acid terephthalic acid and N,N′-dimethylformamide. Porous NPs were designed to entrap vancomycin (Van) drugs. The conclusion of the study brought remarkable results, since NPs acted as a carrier platform for vancomycin-encapsulated antibacterial drugs. In addition to the efficient drug loading capacity, up to 20% by weight, NP-Van exhibited an antibacterial effect with high efficiency (99.3%) without significant cytotoxicity [6].

Several possibilities for the formation of coordination compounds can be created by the combination of carboxylic acids and their deprotonated derivatives (COO^−^) with metal ions, and their applications can be directed to the biological and medicinal area, where it has shown great efficiency [1,4,5,6]. Because of this, the use of a new carboxylic ligand obtained from a natural resin, Na-abietate, offers the opportunity to carry out syntheses of unprecedented complexes, such as VO(IV)-*bis*(abietate), which is presented in this work.

The abietate ligand used in this work was synthesized by Correa et al. [7], by the salification reaction of abietic acid present in pine resin, of the species *Pinus elliottii var. elliottii*. Its molecular formula (C_20_H_29_O_2_)^−^ corresponds to that of its precursor in the deprotonated form and is linked to a sodium atom by the carboxylic function present in the molecule [(C_19_H_29_COO)Na]; therefore, it is called sodium abietate (Na-abietate, Figure 1), it presents predominantly a yellow color, characteristic of the precursor resin.

The metal ion is also relevant for the carboxylate complex, since different metals lead to distinct applications in catalysis, or as a sensor, pigment, etc. In particular, vanadium metal is very interesting due to its variability of oxidation states (+5, +4, +3, +2, and +1) allowing different forms of bonding, which can influence the structural characteristics for catalytic, microbiological, and even medicinal applications [8,9,10,11,12]. An impasse in the use of vanadium is its toxicity, some studies point out that the metal is essential in small amounts and toxic in excess [13]. Its toxicity depends on factors, such as the type of compound (inorganic/organic), nature of the ligands, valence, etc. At low concentrations, vanadium has a beneficial effect on the growth and physiological functions of some microorganisms, plants, and fungi; however, the mechanisms of metal toxicity need further studies to be better elucidated [13].

Currently, the problem that has caused the most concern to the scientific community is the adaptation of disease-causing microorganisms and the resistance developed against traditional drugs. Accordingly, several studies involving the synthesis of new compounds that inhibit the proliferation of bacteria, fungi, and viruses have been carried out [14,15,16,17,18].

Evidence of very positive results involving vanadium complexes has been reported. For instance, Datta et al. [19] cited the oxovanadium(IV) complex [VO(C_16_H_15_N_4_O_8_S)HSO_4_] incorporated into the antibiotic cefuroxime as a potential antioxidant and antibacterial agent. Another five compounds encompassing vanadium oxide, 8-hydroxyquinoline derivatives, and the Schiff base ligand, were also reported and studied in vitro against *Trypanosoma cruzi*, being classified as a potential agent against protozoan [20]. Vanadium is also related to the treatment of diabetes because they are a potent hypoglycemic agent. Studies involving vanadium compounds have already reached the clinical stage, while many others exhibited good effectiveness in vitro studies. The work of Wei and Yang [21] reported the synthesis of *bis*(maltolate)oxovanadium(IV) (BMOV), which has reached clinical phase II in humans. However, toxicity to the kidneys of patients has been identified, and the tests were interrupted. To circumvent the problem, the authors proposed a new vanadium compound, *bis*((5-hydroxy-4-oxo-4H-pyran-2-yl)methyl benzoatate)oxovanadium (IV) (BBOV), to reduce toxicity. The compound managed to restore blood glucose in the test carried out in vivo using rats with induced diabetes; however, there is no record of testing in humans so far [21,22].

According to the literature, the importance of vanadium metal and carboxylate ligands in biological and medicinal areas is remarkable. Therefore, in this work, the green and economically viable synthesis of a vanadium complex with the abietic ligand was carried out and tested in the microbiological area as an antibacterial and antifungal agent, either as an active solution or dispersed in commercial white paint.

## 2. Results

In this work, we report the green synthesis of an oxovanadium(IV)-*bis*(abietate) complex, by a low-cost and efficient method. Firstly, the synthesis of Na-abietate salt (abietate ligand) was carried out from the salification of pine resin, species *Pinus elliotti var. elliotti*. The ligand was synthesized and characterized, as discussed by Correa et al. [7]. The synthesis of the VO(IV)-*bis*(abietate) complex occurred by a simple, direct route and without special conditions of atmosphere or temperature control. After characterization, biological tests were performed on VO(IV)-*bis*(abietate) in solution and on the solid dispersed as an inorganic pigment in paint. The VO(IV)-*bis*(abietate) showed remarkable results in the inhibition of bacteria and yeast.

### 2.1. Chemical Composition (CHN, MS)

The elemental analysis data (CHN) detected the expected percentages for VO(IV)-*bis*(abietate), C = 65.9% and H = 8.3% (C/H_exp_ = 7.94), due to the higher mass fraction of the complex being the organic ligand abietate (C_20_H_29_O_2_^−^; MM = 301.44 g.mol^−1^; C = 79.68% and H = 9.69%; C/H_theorical_ = 8.22).

The mass spectrum (Figure 2) of VO(IV)-*bis*(abietate) was obtained in dichloromethane (DCM) solution diluted in methanol. The most intense peak was observed at m/z 1- 301.28, corresponding to the theoretical molecular mass of deprotonated abietic acid. A second peak was observed at m/z 1- 601.51 relative to the formation of a dimeric form of abietic acid. The third peak that stands out at m/z 1- 683.51 corresponds to the formation of the complex, with the suggested molecular formula [VO(C_20_H_29_O_2_)_2_].

### 2.2. Thermal Analysis (TGA) and Vibrational Spectroscopy (FTIR)

Considering the proposed molecular formula [VO(C_20_H_29_O_2_)_2_], the first stage of mass loss occurs with a peak at 59.09 °C with 3.98% mass loss, attributed to the loss of weakly bound water molecules (*n* = 1.6 molecules) in the compound (Figure 3). The two abietate ligands correspond to 85.8% mass loss, starting at the second to the fifth peak (Figure 3). The total loss of mass was 89.7%, and there was 8.44% of the mass remaining at the end of the analysis, referring to the formation of vanadium oxide, possibly VO_2_. The thermal decomposition of VO(IV)-*bis*(abietate) occurs according to the proposal presented in Table 1.

The infrared spectrum (Figure 4) shows characteristic bands of the formation of the VO(IV)-*bis*(abietate) complex. Figure 4a corresponds to the abietate ligand, and the bands at 1544 cm^−1^ are attributed to ν_asym_(COO^−^), and at 1397 cm^−1^, they are assigned to ν_sym_(COO^−^). These two bands undergo displacement when coordination occurs, indicating the mode of the ligand in relation to the metal ion [1,7].

In the vibrational spectrum for VO(IV)-*bis*(abietate) (Figure 4b), the bands at 2929 cm^−1^ and 2868 cm^−1^ attributed to the C-H stretches stand out. The main bands to be highlighted are at 1693 cm^−1^ and 1640 cm^−1^, attributed to the asymmetric and symmetrical stretching, respectively, of the carboxylate group present in the ligand. These bands are typical of the COO^−^ group but may vary according to the ligand and the metallic center [1]. The characteristic peak of the V=O bond occurs around 970 cm^−1^. Peaks in the 1273 cm^−1^ region may be indicative of C=O binding.

The coordination effect of the abietate ligand is noticeable when comparing the spectra of Na-abietate and VO(IV)-*bis*(abietate) (Figure 4), the bands undergo displacement due to the coordination of the ligand to the vanadium ion. Furthermore, the FTIR spectrum confirms the binding mode caused by the difference between the vibrational stretching modes of the carboxylate group, ν_asym_(COO^−^) and ν_sym_(COO^−^).

From the values of the positions of the bands, the difference can be calculated (Δ = (ν_asym_(COO^−^) − ν_sym_(COO^−^)); Δ = 147 cm^−1^ for Na-abietate; and Δ = 53 cm^−1^ for VO(IV)-*bis*(abietate). This value indicates the presence of possible terminals of bidentate bridges, since according to studies, a value Δ ≤ 200 indicates the presence of bidentate bonds in carboxylate molecules [1,19].

### 2.3. Structural Analysis by XRD

The X-ray diffraction profiles (XRDP) of the samples (Figure 5) were analyzed by diffraction (Bruker AXS 2012). No matching files were found in the Inorganic Crystal Structure Database (ICSD); International Centre for Diffraction Data (ICDD), and Crystallography Open Database (COD).

In the XRDP for Na-abietate, three characteristic peaks were identified at 2θ 12.16° (d = 7.27 Å), at 2θ 15.77° (d = 5.62 Å), and at 2θ 29.99° (d = 2.98 Å). For the VO(IV)-*bis*(abietate), five diffraction peaks are highlighted at 2θ 7.38° (d = 11.97 Å), at 2θ 14.86° (d = 5.95 Å), at 2θ 26.46° (d = 3.37 Å), in 2θ 31.56° (d = 2.83 Å), and in 2θ 45.29° (d = 2.00 Å). Figure 5 shows the similarity between the diffractograms of the two compounds, mainly in relation to the peak at 2θ 15.77° for Na-abietate and at 2θ 14.86° for VO(IV)-*bis*(abietate), characteristic of lamellar material or organization in the form of sheets.

According to the literature [23], and corroborated by the mass spectrum, the complex is oxovanadium(IV)-*bis*(abietate), and in this case, always (without exception) the structure is V=O (square pyramid), so the carboxylates have to be bidentate (FTIR spectrum, Figure 4). The structure of the complex is shown in Figure 2, with the mass spectrum.

The calculation of the degree of crystallinity for Na-abietate was 52% and the size of the crystallite found was 21.9 Å; for VO(IV)-*bis*(abietate) the degree of crystallinity calculated was 78.7% and with a crystallite size of 22.7 Å; both were calculated in the range of degree 2 theta between 7° and 60°, and k factor equal to 1. The highest degree of crystallinity found for VO(IV)-*bis*(abietate) suggests greater organization due to the formation of a well-defined structure complex, according to the mass spectrum data.

### 2.4. Electronic Spectroscopy (Visible-NIR)

Vanadium has four stable oxidation states; V^+5^ is yellow, V^+4^ is dark blue, and V^+3^ is green [12]. Therefore, the formation of the complex follows an oxidative substitution reaction, shown in Equation (1).
(1)$$4V^{+3}Cl_{3\left(aq\right)}+4Na{\left(C_{19}H_{29}COO\right)}_{\left(aq\right)}+O_{2\left(g\right)}+4H_{\left(aq\right)}^{+}+4e^{-}\to 4{\left[V^{+4}{\left(C_{19}H_{29}COO^{-}\right)}_{2}Cl_{2}\right]}_{\left(s\right)}+4NaCl_{\left(s\right)}+2H_{2}O_{\left(l\right)}$$

Figure 6a shows the electron spectra in absorbance and reflectance modes for the VO(IV)-*bis*(abietate) complex in powder form. The absorbance spectrum shows a broad band centered at ~460 nm, attributed to the electronic transition d-d to ion d^1^ to V^+4^ [11,23]. In the reflectance spectrum, the presence of the band centered at 575 nm stands out, characteristic of greenish-yellow samples. The reflectance index decreases along wavelengths between 650–900 nm, characteristic of a warm pigment.

Figure 6b shows the reflectance spectra for VO(IV)-*bis*(abietate) dispersed as a pigment in white paint, at concentrations of 5%; 10%, and 15% (*w*/*w*). For comparison, the reflectance measurement was performed only for the white paint without pigment. As the pigment concentration increases, there is a decrease in the reflectance percentage, this behavior is typical of hot pigments, which absorb NIR radiation, important in the thermal control of internal and external environments. For the paint containing the pigment, a broad band between 500–650 nm is observed, corresponding to the region of cool color green, followed by the components of warm colors yellow and orange. In other words, VO(IV)-*bis*(abietate) presents itself as a pigment with a broad spectrum of absorption and reflectance.

### 2.5. Colorimetric Analysis

The colorimetric analysis was performed by the CIELab color space method [24]. The method allows the discussion of the difference in lightness (L), hue (h), and chroma (C) between pure color samples dispersed in white or colorless paint or other dispersing mediums compatible with pigments or dyes. The parameters L*a*b* represent luminosity (L = 100 for white, and 0 for black), red (a+) and green (a-) colors, and yellow (b+) and blue (b-), as shown in Figure 7.

The graph of Figure 7 shows that the VO(IV)-*bis*(abietate) pigment, and the specimens painted with pigmented white paint, are in the a+ (yellow) and b- (green) quadrant, that is, they have a greenish-yellow hue. The L* values tend towards an increase in luminosity due to dispersion in the white paint (Table 2), but with a decrease in chroma and duller.

### 2.6. Antimicrobial Activity

#### 2.6.1. VO(IV)-*bis*(abietate) Solution

Table 3 presents the results of the antimicrobial tests for the VO(IV)-*bis*(abietate) complex (2 mmol L^−1^). The tests carried out in solution were for the MIC, MBC, MFC, and disk diffusion methods. The VO(IV)-*bis*(abietate) complex in solution had good results for Gram-positive *Staphylococcus aureus* and Gram-negative *Escherichia coli*, both with MIC values of 62.5 µmol L^−1^. For *Listeria monocytogenes*, the concentration was 125 µmol L^−1^, and for *Salmonella enterica* Typhimurium, 1000 µmol L^−1^. The MBC test for VO(IV)-*bis*(abietate) did not have a favorable result; therefore, it has only bacteriostatic activity.

Some studies show vanadium complexes with antibacterial activity. The dioxide-vanadium(V) complex was tested against *S. aureus* and *E. coli*, and showed MIC activity of 250 μg mL^−1^ for both bacteria [14]. In the work by Farzanfar et al. [25], three complexes of oxide-vanadium(IV) and one of dioxo-vanadium(V) with ligands derived from thiourea were synthesized, which had MIC activity from 256 to 512 μg ml^−1^ for *E. coli*, and from 64 to 512 μg ml^−1^ for *S. aureus*. The work that used V(IV) complex shows MIC results from 3125 to 4500 μg ml^−1^ for *S. aureus* and 5500 to 9250 μg ml^−1^ for *E. coli* [26].

In the test with yeast, VO(IV)-*bis*(abietate) had better activity for *Candida albicans* with MIC equal to 15.6 μmol/L and for *Candida tropicalis* MIC equal to 62.5 μmol/L. In this case, it had results for minimum fungicidal concentration (MFC) with a value of 15.6 μmol/L for *C. albicans* and 62.5 μmol/L for *C. tropicalis*. Therefore, for the tested yeast, VO(IV)-*bis*(abietate) has antimicrobial activity.

It is important to emphasize that the satisfactory result for *C. tropicalis*, which is considered one of the main strains that have resistance to antifungal therapy, a fact that causes concern for the high rates of infections caused by this and other strains of non-albicans candida [27,28]. The literature [26] shows efficiency for four complexes of vanadium(IV) with Schiff base ligands, with MIC for *C. albicans* from 6.125 to 14.25 μg/mL.

In the diffusion disc method, tested against bacteria only, there were results in all cases. Table 4 shows the inhibition halos obtained, through the average of the triplicate performed and beside the error value calculated by the data processing software. Data were obtained for the control drug (chloramphenicol) for comparison purposes.

The values are presented in the form of a graph (Figure 8a) for better visualization. The largest halo presented was in the inhibition of the Gram-negative bacterium *E.coli*, with 8.72 ± 0.23 mm. We highlight the inhibition value for *L. monocytogenes* (8.31 ± 0.10 mm), compared to the inhibition value for the antibiotic chloramphenicol, which was 10.2 ± 0.47 mm, a difference of only 1.89 mm, which shows the efficiency that the complex has in inhibiting this bacterial strain. The lowest inhibition values occurred for *S. aureus* with a halo of 7.00 ± 0.25 mm and *S.* Typhimurium with a value of 7.10 ± 0.25 mm.

Figure 9 shows the tests performed on the Petri dishes and the halos inhibited for each bacterial strain. In the images, the letter “c” represents the negative control, where only the solvent used is used (DMSO 6.25%) and the letter “a” represents the positive control, chloramphenicol. For *E. coli*, the positive control was performed on another plate to be able to measure the inhibition halo. Note that for the negative control there is no halo, so DMSO alone does not inhibit the growth of bacteria.

The literature presents very good results regarding the diffusion disk method applied to vanadium complexes [14,25,26,29]. One of the works reports a zone of inhibition of 10.4 to 12.6 mm for *E. coli* and 10.8 to 21.6 mm for *S.aureus* for vanadium complexes synthesized with ligands derived from thiourea [25]. Through the results obtained and the literature cited, the relevance of metal complexes in the inhibition of microorganisms stands out, as well as the need to deepen discussions and continue studies to improve the performance of vanadium complexes.

#### 2.6.2. Dispersion of VO(IV)-*bis*(abietate) in White Paint

Diffusion disk assay was also performed for VO(IV)-*bis*(abietate) dispersed in white paint at a concentration of 5% (m/m). Pigmented paint was applied to filter paper and cut into discs for antimicrobial tests.

Figure 9b presents the data in a graph, where a close antibacterial activity is noticed for all strains tested; in this way, the complex can inhibit Gram-positive and Gram-negative bacteria. Figure 10 shows the inhibition halos obtained by the pigmented ink with VO(IV)-*bis*(abietate) (5%). The negative control used was a disc, painted only with white paint; no halos were identified in this case (Figure 10a–d). The best result was obtained for *E.coli* with a value of 10.88 ± 0.13 mm of inhibition, followed by *S. aureus* with 8.81 ± 0.26 mm; *S. enterica* Typhimurium with 8.13 ± 0.07 mm, and finally *L. monocytogenes* with 7.70 ± 0.12 mm.

For *C. tropicalis*, the yeast strain tested in this case, there was no yeast growth in 48 h, the recommended time for the experiment, so it was not possible to measure the halo of inhibition. However, in 96 h there was the growth of colonies only on the control disk, painted with white paint, as can be seen in Figure 10e.

In a bibliographic survey, no pieces of work were found in the same conditions presented in this work for VO(IV)-*bis*(abietate). However, results are found for Zn-abietate incorporated into the ink, reaching efficiency > 99.9% in the inhibition of *S. aureus* and *E.coli* [1], using a method proposed by JIS (Japanese Industrial Standard).

## 3. Materials and Methods

The reagents were purchased commercially and used without any additional treatment, being: sodium hydroxide (Na(OH), Neon^®^, P.A.), vanadium(III) chloride (VCl_3_.xH_2_O, Sigma-Aldrich, P.A.), Mueller Hinton agar and broth (Kasvi), Sabouraud dextrose broth (TM MEDIA), potato dextrose agar da (Himedia), TTC dye (Sigma-Aldric). The pine resin was collected in nature and was purified according to the process described in the literature [2].

### 3.1. Synthesis of Na-Abietate Ligand

The abietate salt ligand, Na(C_20_H_29_O_2_), was synthesized by adding pine resin and a solution of NaOH (0.75 mol L^−1^) in a beaker. For each gram of resin, 0.15 g of NaOH and 5 mL of ultrapure water are used, and the molar ratio used was 1:1 (m:m). Then, the mixture was homogenized and heated under constant stirring, until the total evaporation of the water resulted in the formation of the yellowish Na-abietate salt. The solid was dried in an oven (70 °C) for approximately 3 h. The solid was macerated and used in the synthesis of M-abietate.

To ensure that the ligand is pure, a solid recrystallization step was performed. An amount of 20 g of the Na-abietate was added to 100 g of water, this solution was left stirring for 12 h, then filtered and dried in a desiccator.

### 3.2. Synthesis of VO(IV)-bis(abietate) Complex

Vanadium chloride (VCl_3_.xH_2_O) was solubilized in 20 mL of ultrapure water (0.3 g; 0.095 mol/L). Separately, the aqueous solution of the binder (1.4 g; 30 mL of water; 0.14 mol/L) was prepared, then the solutions were mixed. The formation of a green precipitate was observed, then the solution was filtered, and the solid was dried in an oven (70 °C for 2 h), pulverized in a mortar and pestle, and passed through a sieve (60 mesh). The synthesis methodologies of the ligand and VO(IV)-*bis*(abietate) complex are shown in Figure 1.

### 3.3. Complex Characterization

The mass spectra (MS) of VO(IV)-*bis*(abietate) were obtained from a solution of dichloromethane (DCM) diluted in methanol injected in a Bruker Amazon Speed ETD equipment, ion trap (MS-MS) with low resolution, in negative ion and ionization by electrospray mode. A drying gas flow of 4 L min^−1^ was used at a temperature of 200 °C, nitrogen as a nebulizer gas under pressure of 7 psi, and a voltage of 4500 V. Elemental CNH analysis was performed to estimate the composition of the samples using an elemental analyzer—PerkinElmer 2400 Series II. Fourier transform infrared spectroscopy (FTIR) analyzes were performed on a PerkinElmer Frontier spectrometer, in the 4000–650 cm^−1^ region. The samples were analyzed in the Eco-ATR attenuated total reflectance acquisition mode, equipped with high capacity ZnSe ATR crystal for analysis of powders, solids, pastes, and liquids. Thermal decomposition analysis was performed on a PerkinElmer thermal analyzer, STA 6000, in simultaneous differential scanning calorimetry (STA/TG-DSC). A heating rate of 10 °C min^−1^, in the temperature range of 30 °C to 1000 °C, with the support of alumina samples, in a nitrogen atmosphere with an average flow rate of 40 mL min^−1^. The energy dispersive x-ray spectroscopy (EDS) was performed coupled to a scanning electron microscopy (SEM) equipment, model TM3000, from Hitachi. The samples were characterized by X-ray diffractometry (XRD), obtained from a Bruker X-ray diffractometer D2Phaser, copper cathode (λ = 1.5418 Å), 30 kV potential, 10 mA current, degree 2 theta between 7° and 60°, and increment of 0.2°/seg. Electronics spectroscopy was performed in an Ocean Optics spectrophotometer, USB 2000 model for solid samples, with a tungsten lamp, between 400–1000 nm, in diffuse reflectance mode. The sample of VO(IV)-*bis*(abietate), in powder form, was evaluated by colorimetry. The coordinates were determined by a portable colorimeter (3 nh, model NR60CP), with a D65 light source. The CIE 1976 L*a*b* colorimetric method was used; in this method L*, is the color lightness (L = 0 for black; and L = 100 for white), a* is the green (−)/red (+) axis, and b* is the blue (−)/yellow (+) axis, as recommended by the Commission Internationale de I’Eclairage (CIE).

### 3.4. Dispersion in Paint

A dispersion test of VO(IV)-*bis*(abietate) in commercial white paint was performed as an inorganic pigment. Pigment concentrations (5, 10, and 15%, *w*/*w*) were prepared and dispersed in 10 g of paint. The pigmented paint was applied in 3 coats, on substrates in the form of plaster blocks, and dried at room temperature, simulating real estate painting. Reflectance measurements were performed in the Vis-NIR spectrophotometer and color measurements on the colorimeter.

### 3.5. Antimicrobial Tests

#### 3.5.1. MIC/MBC/MFC Method

Antimicrobial tests were performed against Gram-positive bacteria *Staphylococcus aureus* (ATCC 25923); *Listeria monocytogenes* (ATCC 19111), Gram-negative *Escherichia coli* (ATCC 25922); and *Salmonella enterica* Typhimurium (ATCC 0028). Additionally, against the *Candida tropicalis* (ATCC 13803) and *Candida albicans* (ATCC 10231) yeasts. Minimal inhibitory concentration (MIC), minimum bactericidal concentration (MBC), and minimum fungicidal concentration (MFC) methods were used to evaluate the activity of VO(IV)-*bis*(abietate) in solution.

The microbial inoculum was adjusted at a concentration of 1.5 × 10^8^ CFU mL^−1^ using a 0.5 McFarland scale. Stock solutions of the compounds were prepared at a concentration of 1 mmol L^−1^ and were dissolved in dimethylsulfoxide (DMSO) and water (the final concentration of DMSO in the inoculum was 6.25%). It should be noted that previous tests have been performed with the solvent DMSO and the chosen concentration is not toxic to microorganisms.

The methodology used for the tests was provided by the Clinical and Laboratory Standards Institute (CLSI) [30,31]. For bacteria, 100 μL of Mueller–Hinton (MH) broth was added to a 96-well plate, and 100 μL of VO(IV)-*bis*(abietate) in DMSO (6.25%) solution was added to wells “A”. After homogenization of the medium, 100 μL of the mixture from well “A” was transferred to well “B” and so on to well “H”, thus, performing serial microdilution. The concentrations tested were 1000, 500, 250, 125, 62.5, 31.2, 15.6, and 7.8 μmol L^−1^. Then, the inoculum of bacterial strains was added to all wells. As a negative control, 100 μL of MH broth and DMSO solvent (6.25%) were used, and the antibiotic chloramphenicol (0.12%) was a positive control. The plate was incubated for 24 h at 37 °C. Then, TTC dye (0.125% 2,3,5-triphenyltetrazolium chloride) was added to all wells and the plate was incubated for another two hours. The dye is added to leave a reddish-pink color in the mixture, indicating the presence of bacteria in the respective wells.

The initial test performed was the minimal inhibitory concentration (MIC); by visually reading the color present in the wells, and after verifying the inhibition of the complex, the minimum bactericidal concentration (MBC) test was performed, indicated by the mixture that inhibited bacterial growth in the MIC test. Transferred to Petri dishes containing MH agar and incubated at 37 °C for 24 h. The interpretation of the MBC was performed by the growth (bacteriostatic) or non-growth (bactericidal) of the samples tested on the plates.

The antifungal test was performed similarly; in this case, Sabouraud dextrose broth is used for yeast cultivation, with the antibiotic fluconazole (10%) as a positive control, and follows the same dilution method mentioned above, resulting in the same dilution concentrations. The incubation time was 48 h (28 °C). For determination of the minimum fungicidal concentration (MFC), the wells that showed a positive result for inhibition are replicated in a Petri dish with a medium of Sabouraud dextrose agar. Subsequently, the plates are incubated (28 °C) for 48 h to evaluate the fungicidal or fungistatic potential of VO(IV)-*bis*(abietate). The evaluation consists of observing the growth of microorganisms on the plate (fungistatic effect) or the non-growth of fungi (fungicidal effect).

#### 3.5.2. Disk-Diffusion Method

Antimicrobial activity was evaluated by the diffusion disk method [16]. In Petri dishes containing MH agar and the bacteria to be tested, 6 mm diameter filter paper disks were placed, and 50 μL of the VO(IV)-*bis*(abietate) solution (2 mmol/L) was dripped onto the disks. For bacteria, a disk with only chloramphenicol antibiotic and incubation (37 °C) for 24 h was used as a positive control. The four strains of bacteria were tested. The inhibition zone (diameter of inhibition halo expressed in mm) was measured using a caliper ruler. The average halo sizes of three independent inhibition tests are indicated.

Antimicrobial tests were carried out on VO(IV)-*bis*(abietate) dispersed in paint, at a concentration of 5%. The dispersion was carried out considering the mass of 14.25 g of paint and 0.25 g of the complex. After being properly homogenized, the pigmented ink was applied in two coats on filter paper. After the paper was dry, 6 mm diameter discs were cut and used for inhibition halo tests, involving bacterial and fungal strains. The test performed is the same as previously mentioned [16]; however, in this case, the disk painted with the paint containing complex was placed in the Petri dish. For control, a disk painted only with white paint was used, for both bacteria and fungi. In the case of yeasts, potato dextrose agar was used for cultivation and incubation (28 °C) for 48 h. The result of the inhibition of microorganisms was observed when there was no growth of colonies near the discs containing the compound; this halo was measured with a caliper, and the higher, the better the antimicrobial activity of the tested compound. Data will be expressed as mean ± standard error. Statistical analyses were performed using GraphPad Prism software version 8.0.

## 4. Conclusions

The new VO(IV)-*bis*(abietate) complex [VO(C_19_H_29_COO)_2_] was obtained by a simple and inexpensive synthesis method. The vanadium ion strongly interacted with the ligand, forming a compound with a structure containing two ligands and a VO group, points to a change in the oxidation state from +3 to +4. The vibrational spectrum shows characteristic bands of the carboxylate group, with the indication of a bidentate bond marked by the difference between the positions of the bands of the respective asymmetric and symmetric stretching modes [Δ(ν_asym_(COO^−^) − ν_sym_(COO^−^) = 53 cm^−1^]. The analysis by colorimetry showed that parameters a* and b* combine for a greenish-yellow color.

Biological tests were promising, for *S. aureus* and *E. coli* bacteria, the MIC value was 62.5 μmol L^−1^, for *L. monocytogenes* 125 μmol L^−1^, and *S. enterica* Typhimurium 1000 μmol L^−1^. The MBC test did not show positive results, thus, it is attributed that the complex has bacteriostatic inhibition capacity for Gram-positive and Gram-negative bacteria. For *C. albicans* and *C. tropicalis* yeasts, the complex showed a fungicidal character, as it inhibited their growth. Among the fungal strains, it is highlighted that *C. tropicalis* has greater resistance to antifungal therapy, hence, the increase in concentration necessary for its inhibition. For the discs painted with the pigmented ink with VO(IV)-*bis*(abietate), the results are quite promising with values of growth inhibition for *L. monocytogenes* (7.70 ± 0.12 mm) and *E. coli* (10.88 ± 0.13 mm). For *C. tropicalis*, it was not possible to measure the halo of inhibition of the compost, because colonies of the fungus only grew after 96 h of incubation on the control disk. Therefore, we can consider the compound promising, as it presented antifungal character both in solution and applied to paint. It is important to note that the concentration tested was 5% of pigment dispersed in the paint, higher concentrations can lead to greater efficiency against microbial agents.

The synthesis of oxovanadium(IV)-*bis*(abietate) proved to be experimentally viable, with total transformation of the amount of vanadium ion into a complex, as it is the limiting agent of the formation reaction. Scaling up to industrial quantities does not require large investments; to produce 50–100 kg/day, you will need mixers and a filtration system, followed by purification, according to the steps shown in Figure 1.

## Data Availability

Not applicable.

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
