# Peer review of "Eco-Friendly Synthesis of an Oxovanadium(IV)-*bis*(abietate) Complex with Antimicrobial Action"

_molecules, 2022, doi:10.3390/molecules27196679_

Round 1
Reviewer 1 Report
The authors present a comprehensive study of the antimicrobial properties of the vanadium-abietate complex. An extensive investigation from synthesis to functional evaluation highlighted relevant results, especially in the context of antimicrobial resistance. Some points need to be reviewed before publication.
1. Gram should be capitalized.
2. Figure 9. Scientific names are always italicized.
3. Antibacterial assays lack control. Is DMSO in this proportion toxic to bacteria and fungi?
4. The authors do not assess and discuss vanadium toxicity.
5. Despite the promising results, some translational limitations and challenges can be discussed in the manuscript.
6. The discussion needs to be enriched.
7. The results presented in Figure 11 have already been presented in Table 5. In this case, it is suggested not to repeat the results in the manuscript. Images of inhibition halos can be placed on supplementary material. On the other hand, a bar indicating the size must be added for a correct interpretation.
Author Response
Dear Reviewer 1,
On behalf of the authors, I thank you for your appreciation and suggestions for correction of the manuscript. The criticisms were essential to improve the final version of the manuscript, contributing to the quality of the text. The answers to the highlighted points are in the attached letter.
Best regards.

Reviewer 2 Report
Anaissi et al. presented the synthesis, characterization and antimicrobial activity of a new vanadium complex incorporating a carboxylate-type ligand coming from the pine resin. Overall, the work looks interesting and may attract the attention of the scientific community working in the field. Before being accepted for publication, the authors should take into considerations the following comments and suggestions for improvements:
- The valence of vanadium, both in the Abstract and text, should be indicated between round brackets.
- In the Abstract, the phrase "Commonly, the synthesis of coordination compounds involves expensive ligands, yielding products." should be written in a clearer manner in order to understand especially the menaing of "yielding products".
- In the Abstract: "imparting a green character"?? A clearer rephrase should be made for a better understanding of the idea.
- Section 2.3 "Structural (XRD) behavior" should be renamed as "Structural analysis by XRD". Although the complex is poorly crystalline and the determination of its real crystal and molecular structure is not possible, the authors should depict a proposed structure of the complex, along with the corresponding structure description.
- From the discussion of XPS and optical spectra of V-abietate complex it is not clear what the oxidation state of vanadium actually is in the powdered sample. This aspect should be better clarified and explained.
- In the phrase the authors state that "at 668 cm-1 and between 2366-2296 cm-1 there are bands of lower intensity that indicate the presence of the -CO2 group." This statement should be better explained in terms of actual identity of -CO2 group.
- Since the Na-abietate ligand was taken from a pine resin, how are the authors sure about its purity? This should be explained.
Author Response
Dear Reviewer 2,
On behalf of the authors, I thank you for your appreciation and suggestions for correction of the manuscript. The criticisms were essential to improve the final version of the manuscript, contributing to the quality of the text. The answers to the highlighted points are in the attached letter.
Best regards.

Round 2
Reviewer 1 Report
All my concerns have been adequately addressed. I thank the authors for their responses. In my opinion, the manuscript have been significantly improved.
Reviewer 2 Report
The authors' responses are satisfactory and their manuscript can now be accepted for publication.